

# Improved isolation and detection of toxigenic *Vibrio parahaemolyticus* from coastal water in Saudi Arabia using immunomagnetic enrichment

Mariam Almejhim[1], Mohammed Aljeldah[2] and Nasreldin Elhadi[1]

[1] Department of Clinical Laboratory Science, College of Applied Medical Sciences, Imam Abdulrahman Bin Faisal University, Dammam, Saudi Arabia
[2] Department of Clinical Laboratory Science, College of Applied Medical Sciences, University of Hafr Al-Batin, Hafr Al-Batin, Saudi Arabia

## ABSTRACT

**Background:** *Vibrio parahaemolyticus* is recognized globally as a cause of foodborne gastroenteritis and its widely disseminated in marine and coastal environment throughout the world. The main aim of this study was conducted to investigate the presence of toxigenic *V. parahaemolyticus* in costal water in the Eastern Province of Saudi Arabia by using immunomagnetic separation (IMS) in combination with chromogenic Vibrio agar medium and PCR targeting *tox*R gene of species level and virulence genes.

**Methods:** A total of 192 seawater samples were collected from five locations and enriched in alkaline peptone water (APW) broth. One-milliliter portion from enriched samples in APW were mixed with an immunomagnetic beads (IMB) coated with specific antibodies against *V. parahaemolyticus* polyvalent K antisera and separated beads with captured bacteria streaked on thiosulfate citrate bile salts sucrose (TCBS) agar and CHROMagar Vibrio (CaV) medium.

**Results:** Of the 192 examined seawater samples, 38 (19.8%) and 44 (22.9%) were positive for *V. parahaemolyticus*, producing green and mauve colonies on TCBS agar and CaV medium, respectively. Among 120 isolates of *V. parahaemolyticus* isolated in this study, 3 (2.5%) and 26 (21.7%) isolates of *V. parahaemolyticus* isolated without and with IMB treatment tested positive for the toxin regulatory (*tox*R) gene, respectively. Screening of the confirmed *tox*R gene-positive isolates revealed that 21 (17.5%) and 3 (2.5%) were positive for the thermostable direct hemolysin (*tdh*) encoding gene in strains isolated with IMB and without IMB treatment, respectively. None of the *V. parahaemolyticus* strains tested positive for the thermostable related hemolysin (*trh*) gene. In this study, we found that the CaV medium has no advantage over TCBS agar if IMB concentration treatment is used during secondary enrichment steps of environmental samples. The enterobacterial repetitive intergenic consensus (ERIC)-PCR DNA fingerprinting analysis revealed high genomic diversity, and 18 strains of *V. parahaemolyticus* were grouped and identified into four identical ERIC clonal group patterns.

**Conclusions:** The presented study reports the first detection of *tdh* producing *V. parahaemolyticus* in coastal water in the Eastern Province of Saudi Arabia.

Corresponding author
Nasreldin Elhadi,
nmohammed@iau.edu.sa

# INTRODUCTION

*Vibrio parahaemolyticus* is a halophilic bacterium that is abundant in marine and estuarine environments (*Kalburge, Whitaker & Boyd, 2014*). The highest abundance of *V. parahaemolyticus* is in sediment and benthic environments (*Böer et al., 2013*; *Alipour, Issazadeh & Soleimani, 2014*). *V. parahaemolyticus* is also present in various types of marine seafood and organisms, such as shrimps, mollusks, oysters, fish, crabs, lobsters, mussels, and zooplankton (*DePaola et al., 2003*; *Su & Liu, 2007*; *Julie et al., 2010*; *Letchumanan, Chan & Lee, 2014*). However, the growth of *V. parahaemolyticus* has an absolute salt requirement for survival and is capable of growth at 1% to 9% NaCl (*Whitaker et al., 2010*; *Kalburge, Whitaker & Boyd, 2014*). The presence of *V. parahaemolyticus* in the environment widely varies according to differences in geographical locations and environmental factors, such as temperature and salinity (*Parveen et al., 2008*; *Johnson et al., 2012*). *V. parahaemolyticus* is reported to grow at temperatures ranges from 5 °C to 43 °C (*ICMSF, 1996*) and the salinity ranges between 1 to 35 parts per thousand (ppt) (*Larsen et al., 2015*; *FAO & WHO, 2020*). Several studies have reported an association between the isolation of *V. parahaemolyticus* and a higher temperature of seawaters (*Blackwell & Oliver, 2008*; *Yoon et al., 2008*). In the Chesapeake Bay, USA, the detection of *V. parahaemolyticus* was rare and difficult until the temperature reached 19 °C or above (*Kaneko & Colwell, 1973*). Also, during the winter season, *V. parahaemolyticus* will survive in the sediment; it usually appears in the water column at the end of spring or the beginning of the summer season (*Julie et al., 2010*). On the other hand, the salinity of the seawater affects the presence or absence of *V. parahaemolyticus* in the environment (*Johnson et al., 2010*).

The most pathogenic virulence factors of *V. parahaemolyticus* are thermostable direct hemolysin (TDH), and TDH-related hemolysin (TRH) (*Tada et al., 1992*), but the underlying mechanism of these proteins in human infection remains unknown (*Broberg, Calder & Orth, 2011*; *Ceccarelli et al., 2013*). *V. parahaemolyticus* can cause wide-scale infection transmitted through the consumption of raw or undercooked contaminated seafood, usually during the warmer months (*Baker-Austin et al., 2017*). The infection takes between 4 to 24 h; then, the symptoms begin to appear and self-resolve within 48 to 72 h. However, three significant medical conditions can be caused by *V. parahaem*olyticus: acute gastroenteritis, wound infection, and septicemia (*Nair et al., 2007*). Acute gastroenteritis appears with abdominal pain, diarrhea, vomiting, nausea, and headache with fever, as well as sometimes with bloody diarrhea (*Li et al., 2016*). Wound infection is commonly detected in fishermen with a small wound occurring at the time of fishing in contaminated seawater; usually, the infected person will suffer from cellulitis, though in some cases, the infection will progress to severe necrotizing fasciitis (*Hlady & Klontz, 1996*). Very few cases of *V. parahaemolyticus* lead to septicemia, which might be fatal to a person with an underlying medical problem, including immunocompromised patients, such as those with cancer or liver diseases (*Jia et al., 2016*).

*V. parahaemolyticus* is commonly isolated from Asian regions because of the nature of the food consumed in these countries. Therefore, outbreaks start in various countries in Asia, such as Japan, India, and China (*Hara-Kudo et al., 2001*; *Yonekita et al., 2020*). The main cause of outbreaks in Asian regions has been reported as the consumption of contaminated seafood (*Jacxsens et al., 2009*; *Yonekita et al., 2020*). Globally and compared with other foodborne illnesses, *V. parahaemolyticus* infections have been increasing and have become the leading cause of seafood bacterial infections (*Martinez-Urtaza et al., 2010*; *Abanto et al., 2020*). In Saudi Arabia, no report of isolation of *V. parahaemolyticus* from any clinical cases and most of the reported isolation from environmental samples (*Elhadi, 2013*; *Elhadi & Nishibuchi, 2018*; *Ghenem & Elhadi, 2018*).

The US Centers for Disease Control and Prevention (CDC) estimated that the average annual incidence of all *Vibrio* infections increased by 54% during 2006–2017 (*Marder et al., 2018*), and *V. parahaemolyticus* was responsible for the highest number of infections (*Newton et al., 2014*). In the United States, *V. parahaemolyticus* is responsible for more than 35,000 human infections per year, and in China, since 1990, *V. parahaemolyticus* has been registered as the leading cause of foodborne infections (*Scallan et al., 2011*; *Liu et al., 2018*). In the summer of 2004, in Alaska, 14 passengers were infected on a cruise trip after consuming raw oysters (*McLaughlin et al., 2005*). The largest outbreak of *V. parahaemolyticus* was in the summer of 2012 on a cruise boat in Spain; 100 out of 114 passengers were infected. After a laboratory investigated, they found different genes and reported the first presence of *V. parahaemolyticus* strains carrying both the *tdh+* and *trh+* pathogenicity genes (*Martinez-Urtaza et al., 2016*).

Not all *V. parahaemolyticus* strains are pathogenic, only those expressing *tdh* that encodes the *tdh* or *trh* genes (*Tada et al., 1992*; *Ceccarelli et al., 2013*; *Saito et al., 2015*). Therefore, the objectives of the present study were (i) to isolate *V. parahaemolyticus* from coastal water by using IMBs in samples treatment to concentrate bacteria after the enrichment process; (ii) to confirm all isolates of *V. parahaemolyticus* to species level by using PCR targeted to the *tox*R gene; (iii) to examine all *tox*R gene-positive isolates for the presence of the *tdh* and *trh* genes using PCR; and (iv) to genotype all isolates of *V. parahaemolyticus tox*R gene-positive isolated from different locations along the coast of Eastern Province of Saudi Arabia for relative genetic similarity by using enterobacterial repetitive intergenic consensus (ERIC)-PCR.

## MATERIALS AND METHODS

### Study design and sample collection

In the present study, a total of 192 sea surface water samples were collected from five different sites along the coast of the Eastern Province of Saudi Arabia between March 2018 and May 2018. All samples were collected in sterile 500 ml screw-cap bottles from: (i) 40 samples from Al aziziyah beach (AZB); (ii) 39 samples from corniche Al-Khobar (KBC); (iii) 38 samples from corniche Al-Khobar front (KBF); (iv) 36 samples from Dammam beach (DMB); and 39 samples from Half-Moon beach (HMF) as illustrated in Fig. 1. During sample collection from each location, the temperature and pH of surface seawater were measured using a Multi-Parameter Water Quality Meter (YSI-50 series;

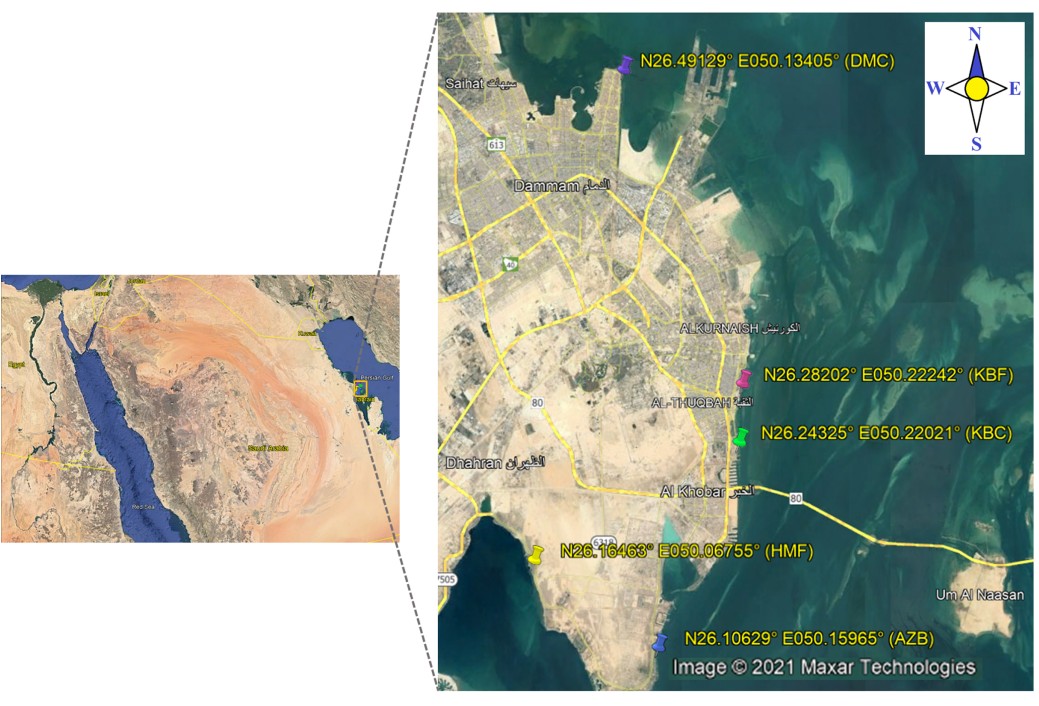

**Figure 1 Map showing locations of the sampling sites.**

Horiba, Irvine, CA, USA). Seawater samples were transported after collection to the microbiology research laboratory at Imam Abdurrahman bin Faisal University (IAU) and processed immediately to test for the presence of *V. parahaemolyticus*. Seawater samples were analyzed in accordance with the FDA method for Vibrio isolation (*DePaola & Kaysner, 2004*). All water samples were enriched in alkaline peptone water (APW). The preparation of 1% and 3% APW was performed as described in the Bacteriological Analytical Manual (*DePaola & Kaysner, 2004*).

### Enrichment process

All samples were treated by adding 25 ml of the seawater sample into 225 ml enrichment medium of alkaline peptone water broth (APW) supplemented with 3% NaCl and incubated at 37 °C for 24 h. On the second day, a loop full of each enriched sample was streaked on Thiosulfate citrate bile salts sucrose (TCBS) agar (Oxoid, Basingstoke, UK) and CHROM agar (CHROM, France) and incubated at 37 °C for 24 h.

### Immunomagnetic beads (IMB) separation of *V. parahaemolyticus*

The concentration of suspected *V. parahaemolyticus* in enriched samples in APW were done using commercially available magnetic beads coated with antibodies against *V. parahaemolyticus* polyvalent K antisera groups I to IX (Denka Seiken, Chuo, Tokyo, Japan). The magnetic bead was prepared as previously described with modifications (*Tanaka et al., 2014*). Briefly, 1 ml from each enriched sample in APW supplemented with 3% NaCl was inoculated into tryptic soy broth (TSB) with 2% NaCl for second enrichment and incubated at 37 °C for 24 h. Then, 1 ml was transferred from the second

**Table 1 Primer used in this study.**

| Primer specificity | Primer sequence | Amplicon size (bp) | Annealing temperature (°C) | Reference |
|---|---|---|---|---|
| toxR | Forward: 5′-GTCTTCTGACGCAATCGTTG-3′ <br> Reverse: 5′-ATACGAGTGGTTGCTGTCATG-3′ | 368 | 63 | (*Kim et al., 1999*) |
| tdh | Forward: 5′-CCACTACCACTCTCATATGC-3′ <br> Reverse: 5′-GGTACTAAATGGCTGACATC-3′ | 251 | 55 | (*Tada et al., 1992*) |
| trh | Forward: 5′-GGCTCAAAATGGTTAAGCG-3′ <br> Reverse: 5′-CATTTCCGCTCTCATATGC-3′ | 250 | 58 | (*Tada et al., 1992*) |

enrichment samples into 1.5 ml tube and mixed with 20 μl of IMB specific to *V. parahaemolyticus*. All mixed tubes with IMB were gently inverted and incubated for 45 min at room temperature. The magnetic concentrator rack was used to separate the beads with captured bacteria from enriched samples and washed three times with phosphate buffer saline (PBS). Finally, the bead-aggregated bacterium were resuspended in 50 μl of PBS and spread on TCBS and CaV agar and incubated at 37 °C for 24 h. At least three to five typical colonies of suspected *V. parahaemolyticus* "sucrose non-fermenting" (green or blue) on TCBS agar and "mauve" colonies on CaV agar were selected from each selective media and sub-cultured on tryptic soya agar (TSA) supplemented with 2%. Then, they were incubated at 37 °C overnight for subsequent PCR confirmation. Finally, the results of isolated *V. parahaemolyticus* with and without IMB treatment could be compared.

## DNA template preparation

DNA extraction was done for all selected colonies on TCBS and CaV agar isolated with and without IMB as described elsewhere (*Elhadi & Nishibuchi, 2018*). Briefly, 1 ml of an overnight test culture in Luria Bertani (LB) broth was transferred into a 1.5 ml tube and centrifuged at 10,000 rpm for 2 min, and the supernatant was discarded. The obtained pellet was suspended in sterilized distilled water and boiled at 100 °C for 15 min. Then the tube was centrifuged at 12,000 rpm for 5 min, and the supernatant was transferred to a new tube and stored at −20 °C until use.

## Identification of *V. parahaemolyticus* to species level using PCR targeted to the *tox*R gene

The confirmation of *V. parahaemolyticus* to species level was performed using PCR targeted to the *tox*R gene, as described previously (*Kim et al., 1999*). All isolates of *V. parahaemolyticus* isolated with and without IMB were screened for *tox*R gene amplicon (size 368 bp) using primer sequence, as indicated in Table 1. Positive and negative DNA controls of *V. parahaemolyticus* strains (ATCC 17802) and *V. alginolyticus* (ATCC 17749) were included in all PCR assays.

A positive *V. parahaemolyticus* (ATCC 17802) and negative of *V. alginolyticus* (ATCC 17749) control were included in each PCR run.

## Detection of virulence gene markers

All *tox*R-positive isolates of *V. parahaemolyticus* recovered on TCBS and CaV agar with and without IMB were tested for the presence or absence of tdh and trh virulence gene markers following a previously described protocol (*Tada et al., 1992*). Briefly, the total volume of the reaction was 25 μl, consisting of 12.5 μl of GoTaq Green Master Mix (Promega, Madison, WI, USA), 2 μl of DNA template, 8.5 μl nuclease-free water (Promega, Madison, WI, USA), and 2 μl of forward and reverse primers (Invitrogen, Chuo, Tokyo, Japan) (Table 1). The positive controls of *V. parahaemolyticus* strains ATCC 17802, *V. parahaemolyticus* AQ3815, and *V. parahaemolyticus* AQ4037 were used in each PCR control for the *tox*R, *tdh*, and *trh* genes, respectively. Amplification of both the 251 and 250 bp region for the *tdh* and *trh* genes were performed following the described conditions by *Tada et al. (1992)*: 35 cycles of denaturation at 94 °C for 1 min, annealing at 59 °C for 1 min, and extension at 72 °C for 1 min, followed by a final extension at 72 °C for 7 min. Finally, 10 μl of amplified products were separated using electrophoresis in 1.5% agarose gels stained with ethidium bromide in 1X Tris borate EDTA buffer (Promega, Madison, WI, USA).

## Molecular typing analysis

In order to study genotypes, *V. parahaemolyticus tox*R positive isolates were fingerprinted using enterobacterial repetitive intergenic consensus (ERIC)-PCR as described elsewhere (*Elhadi, 2018*). Briefly, ERIC-PCR was performed using two repetitive primer set sequences, ERIC1R (5′-ATGTAAGCTCCTGGGGATTCAC-3′) and ERIC2 (5′-AAGTAAGTGACTG GGGTGAGCG-3′), as described previously (*Versalovic, Koeuth & Lupski, 1991*). ERIC-PCR was performed in a volume of 25 μl containing 12.5 μl of GoTaq Green Master Mix (Promega, Madison, WI, USA), 3 μl of DNA template, 2 μl of ERIC primer, and 7.5 μl nuclease-free water. The PCR reactions were performed using a Bio-Rad T100 thermocycler (Bio-Rad, Hercules, CA, USA) as follows: 4 min at 94 °C, followed by 35 cycles of 94 °C for 1 min, 52 °C for 1 min and 65 °C for 1 min, with a final extension at 65 °C for 10 min. The ERC-PCR fingerprint patterns obtained by electrophoresis were analyzed by GelJ software (*Heras et al., 2015*). The dendrogram was constructed with the unweighted average pair group method (UPGMA) with a band position tolerance of 1%.

# RESULTS

## Physical parameters of water

Seawater temperature values ranged from 25 °C to 31 °C during the sampling events from February to May 2018 (Table 2). The highest water temperature was documented during April 2018 at Half-Moon beach (HMF), while the lowest value was logged in February at Alaziziyah beach (AZB). The water pH values ranged from 7.35 to 8.46. The highest pH was recorded during April 2018 in Dammam corniche (DMC), while the lowest was recorded during March 2018 in Alkhubar corniche (KBC) (Table 2). The highest number of positive samples for *V. parahemolyticus* was recorded in Half-Moon beach, and the seawater pH and temperature values were 8.22 and 31 °CM, respectively (Table 2).

**Table 2 Distribution of *V. parahaemolyticus* isolated with and without IMB from different locations.**

| Location | No. of samples | Date of collection | Surface seawater | Number of positive samples (%) | | | | |
| | | | | Isolated without IMB on | | Isolated with IMB on | | |
| | | | Temp (°C) | pH | TCBS agar | CaV | TCBS agar | CaV |
|---|---|---|---|---|---|---|---|---|
| AZB | 40 | 4 March 2018 | 25 | 8.18 | 3 (7.5) | 5 (12.5) | 11(27.5) | 15 (37.5) |
| KBC | 39 | 25 March 2018 | 27.5 | 7.35 | 0 | 0 | 6 (15.4) | 6 (15.4) |
| KBF | 38 | 8 April 2018 | 28 | 8.18 | 0 | 0 | 2 (5.3) | 2 (5.3) |
| HMF | 39 | 22 April 2018 | 31 | 8.22 | 0 | 0 | 17 (43.6) | 19 (48.7) |
| DMC | 36 | 5 May 2018 | 28 | 8.46 | 0 | 0 | 2 (5.6) | 2 (5.6) |
| TOTAL | 192 | | | | 3 (1.6) | 5 (2.6) | 38 (19.8) | 44 (22.9) |

## Immunomagnetic bead (IMB) separation of *V. parahaemolyticus*

In this study, samples were considered positive for *V. parahaemolyticus* based on the preliminary appearance of green and mauve colonies on TCBS and CaV agar isolated with IMB and without IMB enrichment treatment (File S1). Among the examined samples, the highest number of positive samples for *V. parahaemolyticus* was detected in samples enriched with IMB and the lowest in samples enriched without IMB (Table 2). The highest distribution rate of positive samples for *V. parahaemolyticus* isolated with IMB on CaV medium was found at HMF 19 (48.7%) and AZB beach 15 (37.5%) as shown in Table 2.Whereas the highest positive samples rates for *V. parahaemolyticus* isolated with IMB on TCBS agar was 17 (43.7%) and 11 (27.5%) from HMF and AZB beaches, respectively (Table 1). Among the total of 192 seawater samples that were enriched in alkaline peptone water broth and processed without using IMB in secondary enrichment, only 3 (1.6%) and 5 (2.6%) samples were reported positive for *V. parahaemolyticus* on TCBS agar and CHROMagar Vibrio, respectively (Table 2). After enrichment with IMB, the rate number of positive samples was 38 (19.8%) and 44 (22.9%) for *V. parahaemolyticus* on TCBS and CaV agar from all locations, respectively (Table 2). A total of 48 and 58 isolates of *V. parahaemolyticus* were isolated on TCBS and CaV agar with IMB, respectively (Table 2). Among the five locations, the highest number of *V. parahaemolyticus* isolates were isolated from HMF (Table 2). The abundance of *V. parahaemolyticus* isolates were recovered on TCBS and CaV agar from samples examined from all locations after using IMB in the secondary enrichment process (Table 2). In this, we found that the use of IMB in secondary enrichment of examined seawater samples could successfully recover typical colonies of *V. parahaemolyticus* on CaV and TCBS agar, outperforming when samples were plated on CaV and TCBS agar without the use of IMB in secondary enrichment.

## Confirmation of *V. parahaemolyticus* to species level using PCR targeted to the *tox*R gene

To confirm the identification of *V. parahaemolyticus* more accurately to species level and for comparison purposes, a total of 120 isolates of *V. parahaemolyticus* isolated using IMB were subjected to PCR with species-specific primers (Table 1). Among 120 isolates of

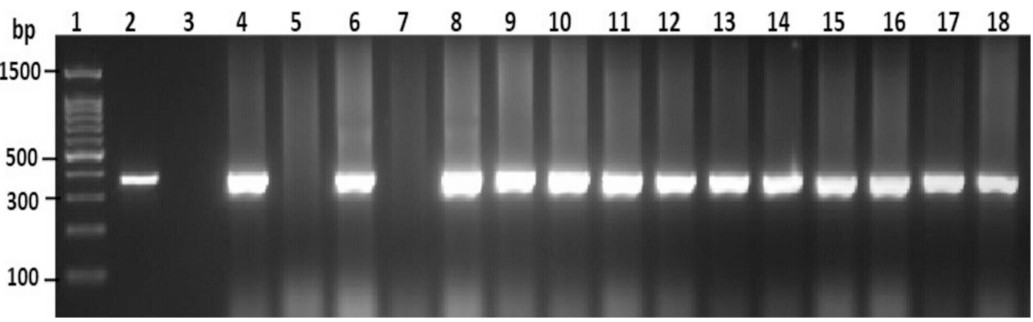

**Figure 2 Representative electrophoresis image of PCR amplification of the *tox*R gene (368 bp fragment).** Lane 1, molecular weight marker (100 bp DNA ladder; Promega); 2, *V. parahaemolyticus* ATCC 17802 (positive control); 3, negative control; 4–18, tested isolates for *tox*R gene.

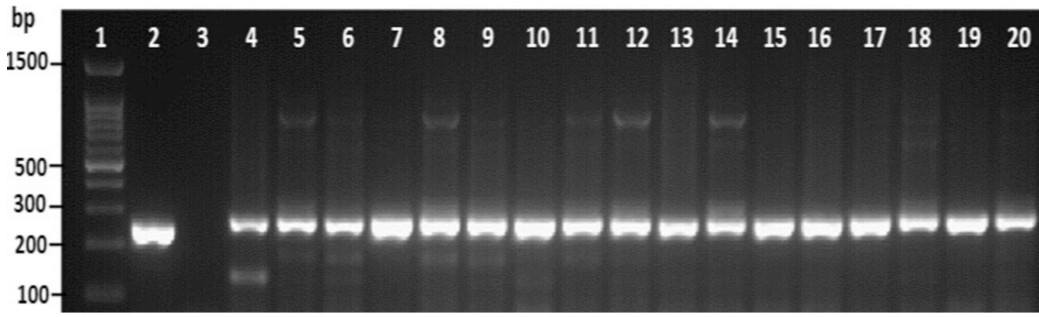

**Figure 3 Representative electrophoresis image of PCR amplification of the *tdh* gene (251 bp fragment).** Lane 1, molecular weight marker (100 bp DNA ladder; Promega); 2, *V. parahaemolyticus* AQ3815 (positive control); 3, negative control; 4–20, positive isolates for *tdh* gene isolated in this study.

*V. parahaemolyticus*, 3 (2.5%) and 26 (21.7%) isolates of *V. parahaemolyticus* isolated without and with IMB were positive for the toxR gene as judged by amplification of a 368 bp fragment (Fig. 2 and File S2).

## Detection of virulence gene markers

Among the overall *tox*R gene-positive *V. parahaemolyticus*, 24 (20%) amplified the 251 bp *tdh* fragment and the highest number of tdh positive isolates was detected from HMF (Fig. 3 and File S3). As presented in Table 3, 21 (17.5%) of the *tdh* gene-positive isolates of *V. parahaemolyticus* were isolated with IMB enrichment, and only 3 (2.5%) of the *tdh* gene-positive isolates of *V. parahaemolyticus* were isolated without IMB enrichment. None of the *V. parahaemolyticus* isolates tested positive for the *trh* gene (Table 3).

## Molecular typing

Among 29 isolates of *tox*R gene-positive *V. parahaemolyticus*, 24 isolates were genotyped using ERIC-PCR DNA fingerprinting analysis and generated high genomic diversity among *V. parahaemolyticus* isolates (File S4). The ERIC primer sets produced 4 to 10 fingerprint bands and ranged between 100 to 1,200 bp (Fig. 4). Of the 24 genotyped isolates of *V. parahaemolyticus*, 18 isolates were grouped and identified into four ERIC identical

**Table 3 Confirmation of *V. parahaemolyticus* isolated with and without IMB treatment targeting species (*tox*R) and virulence gene (*tdh/trh*) markers by using PCR.**

| Location | No. of tested isolates | Number of *V. parahaemolyticus* isolates | | | | | |
| --- | --- | --- | --- | --- | --- | --- | --- |
| | | Isolated without IMB and positive for: | | | Isolated with IMB and positive for: | | |
| | | *tox*R gene | *tdh* gene | *trh* gene | *tox*R gene | *tdh* gene | *trh* gene |
| AZB | 45 | 3 | 3 | 0 | 4 | 3 | 0 |
| KBC | 17 | 0 | 0 | 0 | 1 | 1 | 0 |
| KBF | 6 | 0 | 0 | 0 | 0 | 0 | 0 |
| HMF | 48 | 0 | 0 | 0 | 19 | 15 | 0 |
| DMC | 4 | 0 | 0 | 0 | 2 | 2 | 0 |
| Total (%) | 120 | 3 (2.5%) | 3 (2.5%) | 0 | 26 (21.7%) | 21 (17.5%) | 0 |

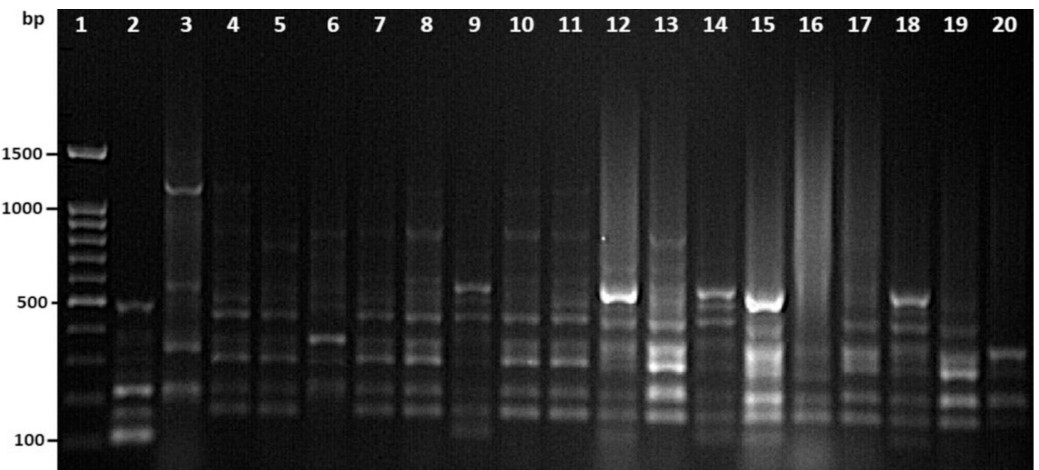

**Figure 4 Amplified DNA fingerprints produced by ERIC-PCR.** Lane 1, 100 bp DNA ladder; from lane 2 to 20 representative isolates of *V. parahaemolyticus* isolated in this study.

clonal group patterns (Cluster 1, 2, 3, and 4), while a similarity cutoff value of 100% was applied and six isolates have shown a single cluster (SC) (Table 4 and Fig. 5). Among the four clusters, cluster-two was comprised of the highest number of *V. parahaemolyticus* isolates with identical clonal origin isolated from AZB, HMF, and KBC corniche between March and April 2018 (Fig. 5).

## DISCUSSION

In this study, potentially toxigenic *V. parahaemolyticus* was isolated from all samples locations, except seawater samples were collected from Alkhubar corniche (KBC). The occurrence of *V. parahaemolyticus* is usually associated with temperature, especially high temperate climate (*Di et al., 2017*). In this study, the highest reported temperature during sampling events was 31 °C and recorded in Half-Moon beach (HMF) during month of April (Table 2). The minimum and maximum pH values for the growth of *V. parahaemolyticus* in the environment were reported to be 4.8 and 11, respectively

**Table 4 Molecular characterization of _V. parahaemolyticus_ isolated with and without IMB treatment.**

| No | Strain code | Sample location | Isolation date | _tox_R gene | Virulence gene | | ERIC type |
|----|-------------|-----------------|----------------|-------------|------|------|-----------|
| | | | | | _tdh_ | _trh_ | |
| 1 | VP-7 | AZB | 4 March 2018 | + | + | − | ET-4 |
| 2 | VP-31-A | AZB | 4 March 2018 | + | + | − | ND* |
| 3 | VP-31-B | AZB | 4 March 2018 | + | − | − | ET-2 |
| 4 | VP-32 | AZB | 4 March 2018 | + | + | − | ET-2 |
| 5 | VP-36 | AZB | 4 March 2018 | + | + | − | SC$^\Psi$ |
| 6 | VP-37 | AZB | 4 March 2018 | + | + | − | ET-2 |
| 7 | VP-49 | KBC | 25 March 2018 | + | + | − | ET-2 |
| 8 | VP-118 | HMF | 22 April 2018 | + | + | − | ET-4 |
| 9 | VP-121 | HMF | 22 April 2018 | + | + | − | ET-2 |
| 10 | VP-123-A | HMF | 22 April 2018 | + | - | − | ET-2 |
| 11 | VP-123-B | HMF | 22 April 2018 | + | + | − | ET-4 |
| 12 | VP-123-C | HMF | 22 April 2018 | + | + | − | ET-2 |
| 13 | VP-125-A | HMF | 22 April 2018 | + | - | − | SC |
| 14 | VP-125-A | HMF | 22 April 2018 | + | + | − | ET-4 |
| 15 | VP-126 | HMF | 22 April 2018 | + | + | − | ND |
| 16 | VP-129-A | HMF | 22 April 2018 | + | + | − | ET-3 |
| 17 | VP-129-B | HMF | 22 April 2018 | + | − | − | ET-4 |
| 18 | VP-130-A | HMF | 22 April 2018 | + | − | − | ET-3 |
| 19 | VP-130-A-B | HMF | 22 April 2018 | + | + | − | SC |
| 20 | VP-132 | HMF | 22 April 2018 | + | + | − | SC |
| 21 | VP-133 | HMF | 22 April 2018 | + | + | − | ND |
| 22 | VP-134 | HMF | 22 April 2018 | + | + | − | ET-1 |
| 23 | VP-137 | HMF | 22 April 2018 | + | + | − | ET-1 |
| 24 | VP-145 | HMF | 22 April 2018 | + | + | − | ET-1 |
| 25 | VP-151 | HMF | 22 April 2018 | + | + | − | ET-1 |
| 26 | VP-152 | HMF | 22 April 2018 | + | + | − | SC |
| 27 | VP-153 | HMF | 22 April 2018 | + | + | − | ND |
| 28 | VP-165 | DMC | 5 May 2018 | + | + | − | SC |
| 29 | VP-166 | DMC | 5 May 2018 | + | + | − | ND |

Note:
ND*, not determined; SC$^\Psi$, a single cluster.

(_Food & Drug Administration, 2020_). The reported pH in this study are within the optimum range of pH, between 5 and 8.6 (_Whitaker et al., 2010_; _Mudoh et al., 2014_). Effectual methods for isolation and identification of _V. parahaemolyticus_ from clinical, food, and environmental samples are required to speed identification and minimize the risk of infection (_Canizalez-Roman et al., 2011_). Our study was able to confirm the isolation of _V. parahaemolyticus_ in all examined seawater samples from all locations along the coast while using IMB in the secondary enrichment process (Table 2). To achieve these results, two different enrichment in APW broth without and with IMB were spread on

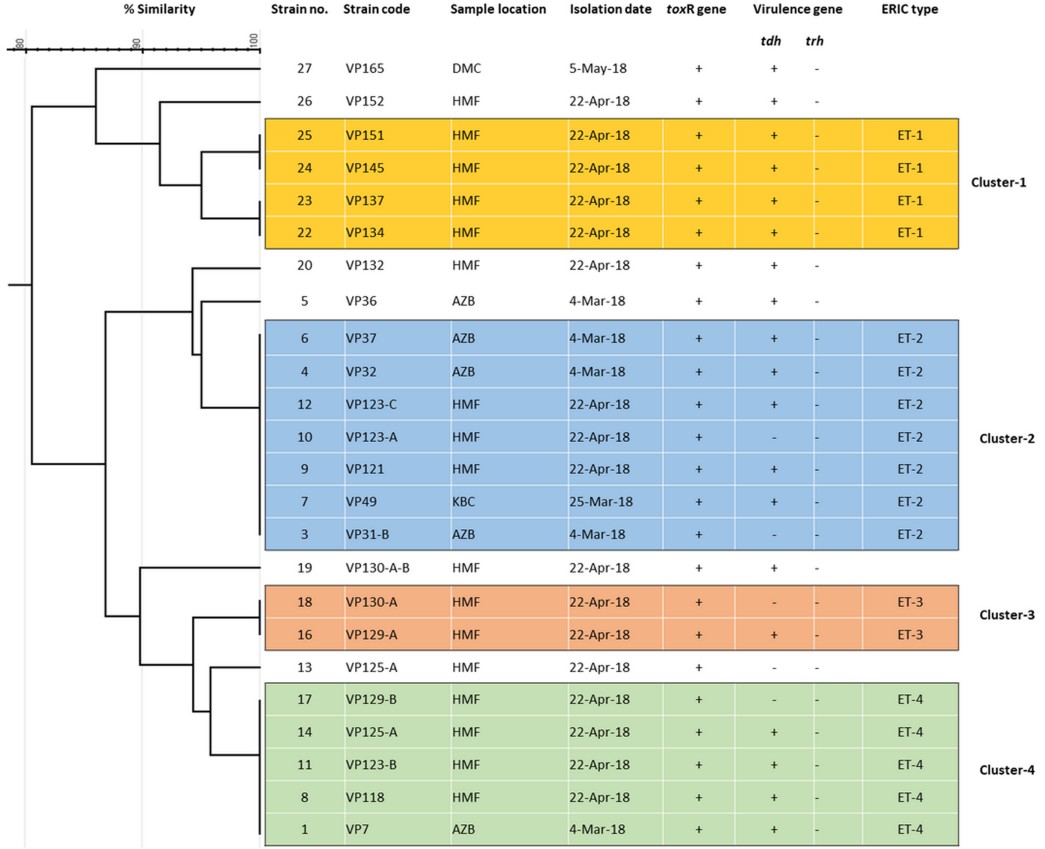

**Figure 5 ERIC-PCR dendrogram of *tox*R and *tdh* gene-positive strains of *V. parahaemolyticus* isolated from coastal water in this study.** Cluster one to four denotes identical clonal groups of *V. parahaemolyticus* strains.

CaV and TCBS agar for isolation of *V. parahaemolyticus*. In this study, we found the use of IMB in secondary enrichment increased the number of positive samples for *V. parahaemolyticus* while using both selective media (Table 2). However, the number of positive samples detected with the use of IMB in secondary enrichment on TCBS agar and CaV medium was 38 (19.8%) and 44 (22.9%), respectively (Table 2). The number positive for *V. parahaemolyticus* without using IMB in secondary enrichment was low in TCBS agar, and CaV medium was 3 (1.6%) and 5 (2.6%), respectively (Table 2).

Consequently, our study used PCR amplification targeted to the *toxR* gene, by which the identity of 3 (2.5%) and 21 (21.7%) isolates of *V. parahaemolyticus* isolated without and with IMB on TCBS agar and CHROMagar Vibrio was confirmed to the species level, respectively (Table 3). Therefore, our study found that the coupling of IMB in secondary enrichment of environmental samples with *toxR* PCR assay is a reliable method for the detection of *V. parahaemolyticus* (*Kim et al., 1999*). Indicators of the potential pathogenicity of *V. parahaemolyticus* is the presence of *tdh* and *trh* genes. Almost all clinically isolated strains of *V. parahaemolyticus* possess hemolytic activity attributed to these two genes (*Ceccarelli et al., 2013*). The effect of TDH on intestinal and epithelial cells is crucial for the biological activities, like diarrhea, during *V. parahaemolyticus* infection

(*Shimohata & Takahashi, 2010*). Also, *trh* works in an analogous pattern to TDH (*Raghunath, 2015*). To the best of our knowledge, this study represents the first report of the detection of *tdh*-positive *V. parahaemolyticus* strains from the coastal environment in the Eastern Province of Saudi Arabia.

The highest percentage of *tdh* positivity was 17.5% from the total isolates detected among *V. parahemolyticus* isolated with IMB enrichment, while the lowest percentage of *tdh* positive isolates was (2.5%) detected in *V. parahaemolyticus* isolated without IMB enrichment (Table 3). From our previous studies were conducted to investigate the presence of toxigenic *V. parahaemolyticus* in costal environment of the Eastern Province of Saudi Arabia, revealed none or less than 1% detection rate of isolates positive for virulence gene markers (*Elhadi & Nishibuchi, 2018*; *Ghenem & Elhadi, 2018*). The detection of TDH positive strains of *V. parahaemolyticus* in the costal environment of the Eastern Province of Saudi Arabia is a pressing concern that has several impacts and requires instant attention. First, the fact that these strains are potentially toxigenic should prompt the healthcare facilities to monitor all bacterial gastroenteritis in clinical samples for the presence of *V.* parahaemolyticus (*Jun et al., 2012*). Second, incidences where pathogenic *V. parahaemolyticus* was held responsible for contaminating seafood produce and not only causing outbreaks of the infection, but also costing the seafood industry enormous economic losses have been documented (*Fuenzalida et al., 2006*; *Thongjun et al., 2013*; *Johnson et al., 2010*). Thus, the results of this study emphasize the continuous monitoring of seafood products' safety. The ERIC-PCR clusters indicate that the isolates could have originated from the same clonal lineage of *V. parahaemolyticus*. These results agreed with our previous study (*Elhadi & Nishibuchi, 2018*) and are consistent with *Marshall et al. (1999)*, who reported that ERIC-PCR was a useful method for evaluating genetic and epidemiological relationships among *V. parahaemolyticus*. The results obtained in this study are very significant for public health in this coastal region and should prompt us to pay attention to the role of these tdh positive *V. parahaemolyticus* in local foodborne diseases. Furthermore, the results should also underline the need for adequate consumer protection against toxigenic *V. parahaemolyticus* in Eastern Province of Saudi Arabia. In particular, setup of a more elaborate protocol is required while dealing with environmental isolates and seafood consumer education regarding proper cooking is considered important. The isolation of toxigenic *V. parahaemolyticus* in coastal water and seafood samples has been well documented in the USA, Japan, Southeast Asia, and many European countries (*Di Pinto et al., 2011*; *Fabbro, Cataletto & Del Negro, 2010*; *Hara-Kudo et al., 2003*; *Johnson et al., 2012*; *Jun et al., 2012*; *Robert-Pillot et al., 2004*; *Roque et al., 2009*). A future study on the prevalence of toxigenic *V. parahaemolyticus* in seafood will be necessary to evaluate public health significance of these strains in the coastal environment of the Eastern Province of Saudi Arabia.

## CONCLUSIONS

The study concluded that, both TCBS and CHROMagar Vibrio are suitable selective media for isolation of *V. parahaemolyticus* if IMBs are used in the enrichment process of

environmental water samples. Therefore, the use of IMB will separate *V. parahaemolyticus* from Vibrio and other non-Vibrio species in environmental samples and improve the isolation level of *V. parahaemolyticus*. This study also concludes that CHROMagar Vibrio has no advantage over TCBS agar if the enriched sample is treated with IMBs coated with specific polyvalent K antisera antibodies for immuno-concentration of *V. parahaemolyticus*. The isolation of tdh positive *V. parahaemolyticus* in this study identifies a public health risk and indicates there is a possibility of the spreading of this gene in the marine environment.

## ACKNOWLEDGEMENTS

The authors would like to dedicate this work to Emeritus Professor Mitsuaki Nishibuchi, the past faculty member of the Center of Southeast Asian Studies, Kyoto University, Kyoto, Japan for useful information and support.

### Funding
The project is funded by the National Science, Technology, and Innovation Programs (NSTIP)-King AbdulAziz City for Science and Technology (KACST), Kingdom of Saudi Arabia, grant number 10-ENV1337-46. The funders had no role in study design, data collection and analysis, decision to publish, or preparation of the manuscript.

### Grant Disclosures
The following grant information was disclosed by the authors:
King AbdulAziz City for Science and Technology (KACST), Kingdom of Saudi Arabia: 10-ENV1337-46.

### Competing Interests
The authors declare that they have no competing interests.

### Author Contributions
- Mariam Almejhim performed the experiments, prepared figures and/or tables, authored or reviewed drafts of the paper, and approved the final draft.
- Mohammed Aljeldah analyzed the data, prepared figures and/or tables, authored or reviewed drafts of the paper, and approved the final draft.
- Nasreldin Elhadi conceived and designed the experiments, performed the experiments, analyzed the data, prepared figures and/or tables, authored or reviewed drafts of the paper, and approved the final draft.

### Data Availability
  The raw data is in the Supplemental Files.

## Supplemental Information

Supplemental information for this article can be found online at http://dx.doi.org/10.7717/peerj.12402#supplemental-information.

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
