# Peer review of "Improved isolation and detection of toxigenic Vibrio parahaemolyticus from coastal water in Saudi Arabia using immunomagnetic enrichment"

_PeerJ, doi:10.7717/peerj.12402_

## Round 0.1 · original submission · Major Revisions

As suggested by reviewers, I recommend revising your paper.

·

Basic reporting

The manuscript is written with relatively good command of English. However, there are some grammatical errors in the manuscript. It includes relevant literature and presented in an organised manner. The authors included clear figures and tables shared in the manuscript.

Experimental design

The methods referred and used in the study, are well-cited. The description of methods are presented with sufficient information for future application.

Validity of the findings

The findings are presented in a clear and systematic manner. The authors has made a comprehensive discussion on robust data gained through their study.

Additional comments

Overall, the manuscript is good. However, the authors could provide some future recommendations on future works in the conclusion.

Reviewer 2 ·

Basic reporting

General comments:
This paper focuses on the isolation and identification of toxigenic Vibrio parahaemolyticus from the coastal water in the Eastern Province of Saudi Arabia. This research is original, and the results are important.
The current manuscript’s writing is clear, and the research is novel. Based on the current manuscript quality, this manuscript should be accepted with minor revisions.

Introduction:
Generally, the goal of the study is clarified. There is sufficient background information about this research and the background information is well connected to the current study. The introduction could add more background knowledge related to this research.

Line 54-55 and Line 62-63: The authors give some basic backgrounds of Vibrio parahaemolyticus – it is a halophile and can survive a wide range of temperature and salinity. I suggest authors writing more details about this bacterium’s physiology. For example, the temperature ranges from XXX degrees to YYY degrees and the salinity ranges XXX to YYY, so the readers will have a clearer picture.

Line 93-105: From “ The US ….” to “ trh+ pathogenicity genes”. I suggest authors put all of these sentences into one individual paragraph, since all of these discuss the examples in the US.

Experimental design

Methods:

Generally, more information about water sample collection is needed.
Line 118: “a total of 192 sea surface water samples were collected”. It is not clear how these water samples were collected. Did the authors filter the water before measuring the pH? Did the authors rinse the tubes before collections or were these tubes sterilized? If the author used a standard protocol to collect water samples, I suggest authors citing this paper below. It used the standard USGS (United States Geological Survey) protocol to collect water samples.
Jiang, X., and Takacs-Vesbach, C.D. (2017) Microbial community analysis of pH 4 thermal springs in Yellowstone National Park. Extremophiles 21: 135-152.

Line 130: It is not clear about APW broth’s recipe. If this is the authors’ own recipe, the ingredients should be listed. If not, a citation should be given here.
Line 177-178: This sentence should be revised and made clearer about the purpose of finger printing and Eric-PCR.
For example, “In order to identify/study the presence of XXX, V. parahaemolyticus toxR positive isolates were fingerprinted using enterobacterial repetitive intergenic consensus (ERIC)-PCR as described elsewhere”.

Validity of the findings

Results:
The authors detailed research results and the results are clear.

Line 201: Revise to “were 8.22 and 31°C, respectively (Table 2).”

Discussion:
Some suggestions on the discussion:

Authors may discuss more at the end of the discussion to make their discussion more robust. For example, what is the significance of current results? Should we be aware of the potential food contamination of Vibrio parahaemolyticus, if the toxic genes are widely dispersed? Based on current research, should normal people be aware of this foodborne pathogen in the Eastern Province of Saudi Arabia? Do the authors have any suggestions and advice on the future study of Vibrio parahaemolyticus.

Line 263-263: “with several studies conducted worldwide that have reported the….”. This sentence is too long to express clear ideas. Revision is needed.
Line 281-283: “disagrees with findings on the unsatisfactory performance of TCBS agar in selecting …….”. It should mention more details about the previous research results here, so readers can understand the differences between current research results and previous results.

Tables and Figures:
In this study, 192 seawater samples were collected from 5 locations in the Eastern Province of Saudi Arabia. I suggested a basic map should be included as a figure. On the map, 5 sample locations should be marked, so it is easy for readers who are not familiar with the geography of Saudi Arabia.

Reviewer 3 ·

Basic reporting

In this study, the author isolated tdh positive V. parahaemolyticus from sea surface water of the coast of the Eastern Province of Saudi Arabia by cultural, IMB, and molecular techniques. They first reported genetically diverse pathogenic V. parahaemolyticus from the coastal water in the Eastern Province of Saudi Arabia. The study was conducted in a short time-period and failed to provide supportive approaches to build up the study. Extensive revision is required in the method and writing for clarification and explanation. The English writing of the work needs to be improved. Some of the sentence structures need to be improved as they are not clear enough to understand. Specifically, lines 25-27, 129-131, 160-161, 302-306, 310-313 are needed to be revised. In this study, the authors mainly compared the isolation of Vibrio parahaemolyticus using IMB on TCBS and CaV agar but the title used here is ‘Isolation and molecular characterization of toxigenic Vibrio parahaemolyticus from the coastal water in the Eastern Province of Saudi Arabia’ which does not match with the study.

Experimental design

The method section needs improvement particularly the sample collection and isolates selection.
Please clarify how the V. parahaemolyticus isolates number was determined? Are multiple isolates taken from the same sample?
Please comment why did you choose one particular month of sample collection instead of throughout the year.

Validity of the findings

In the environment, the prevalence of tdh gene in V. parahaemolyticus is less than 5%. But in this study, 80.77% (21/26) of the toxR confirmed V. parahaemolyticus are positive for tdh gene, which is quite abnormal.

Additional comments

In this study, the author isolated tdh positive V. parahaemolyticus from sea surface water of the coast of the Eastern Province of Saudi Arabia by cultural, IMB, and molecular techniques. They first reported genetically diverse pathogenic V. parahaemolyticus from the coastal water in the Eastern Province of Saudi Arabia. The study was conducted in a short time-period and failed to provide supportive approaches to build up the study. Extensive revision is required in the method and writing for clarification and explanation. The English writing of the work needs to be improved. Some of the sentence structures need to be improved as they are not clear enough to understand. Specifically, lines 25-27, 129-131, 160-161, 302-306, 310-313 are needed to be revised. In this study, the authors mainly compared the isolation of Vibrio parahaemolyticus using IMB on TCBS and CaV agar but the title used here is ‘Isolation and molecular characterization of toxigenic Vibrio parahaemolyticus from the coastal water in the Eastern Province of Saudi Arabia’ which does not match with the study.
Table 3
Primers targeted for the toxR gene are well studied for V. parahaemolyticus species identification, but in this study, only 2.5% (Isolated without IMB) and 26% (Isolated with IMB) of the isolates is confirmed as V. parahaemolyticus. So misidentification of the 120 V. parahaemolyticus may occur here.
Line 39: “17.5% and 2.5% are positive for thermostable direct hemolysin……..” please mention the number of isolates along with the percentage. Are they from 3 and 26 isolates or 120 isolates? If 120, then looking for tdh among them is not logical as they are not even V. parahaemolyticus.

In the environment, the prevalence of tdh gene in V. parahaemolyticus is less than 5%. But in this study, 80.77% (21/26) of the toxR confirmed V. parahaemolyticus are positive for tdh gene, which is quite abnormal.
Introduction:
Please mention if there are some reports of V. parahaemolyticus incidence in the middle east or Saudi Arabia. And the relevance to conducting this study in this area particularly.
Method:
The method section needs improvement particularly the sample collection and isolates selection.
Please clarify how the V. parahaemolyticus isolates number was determined? Are multiple isolates taken from the same sample?
Please comment why did you choose one particular month of sample collection instead of throughout the year.
Line 152: “supernatant was discarded”

Discussion:
Line 263-265: Mentions some references for the previous reports

Please use standard --%(--/--) format in line 34, 39, 210-212, 215,217, 234, 245, 278,280, 287 to make the results clear to the reader.

---

## Round 0.2 · accepted · Accept

Your manuscript was improved after revision and can be accepted for publication.

Reviewer 2 ·

Basic reporting

Dear Editor,

The manuscript has been significantly improved after the revision. The current manuscript’s writing is clear, and the research is novel. The authors have addressed all of questions from my first review. Therefore, I recommend the Editor to accept this manuscript.

Experimental design

The authors have addressed all of questions from my first review.

Validity of the findings

The authors have addressed all of questions from my first review.